# Prolonged Indwelling Urethral Catheterization as Minimally Invasive Approach for Definitive Treatment of Posterior Urethral Valves in Unstable Premature Babies

**DOI:** 10.3390/children8050408

**Published:** 2021-05-18

**Authors:** Silvia Ceccanti, Daniela Pepino, Antonella Giancotti, Ester Ricci, Silvia Piacenti, Denis A. Cozzi

**Affiliations:** 1Pediatric Surgery Unit, Sapienza University of Rome, AOU Policlinico Umberto I, 00161 Rome, Italy; silvia.ceccanti@uniroma1.it (S.C.); esterricci92@gmail.com (E.R.); silvia.piacenti@uniroma1.it (S.P.); 2Pediatric Radiology Unit, Sapienza University of Rome, AOU Policlinico Umberto I, 00161 Rome, Italy; danpepino@hotmail.com; 3Prenatal Diagnosis Unit, Sapienza University of Rome, AOU Policlinico Umberto I, 00161 Rome, Italy; antonella.giancotti@uniroma1.it

**Keywords:** lower urinary tract obstruction, non-operative treatment, indwelling catheter, urinary ascites

## Abstract

Premature newborns with posterior urethral valves (PUV) may present with medical conditions taking priority over definitive surgical care. We encountered three of such cases who underwent initial bladder decompression via transurethral catheterization and waited 2–3 weeks until they were fit enough for voiding cysto-urethrography to confirm PUV. An unexpected good urinary flow and negligible residual urine volume were documented during micturition, suggestive of valve disruption induced by insertion and prolonged duration of indwelling urethral catheter drainage. Cystoscopy documented non-obstructing remnant leaflets. Non-operative treatment may be considered as a viable alternative therapeutic option for PUV in tiny babies facing prolonged intensive care unit stay.

## 1. Introduction

Posterior urethral valves (PUV) are a rare congenital malformation occurring in 1 in every 5000–8000 pregnancies, and represents the most common cause of lower urinary tract obstruction in the newborn male. This malformation is characterized by thin, obstructing membranous folds within the lumen of the posterior urethra. Notably, the affected population carries an almost 30% lifetime risk of end stage renal disease [1]. Such an unfavorable renal outcome is strictly related to the variable degrees of renal dysplasia that is associated with PUV. Additionally, PUV are often associated with a varying degree of oligohydramnios as a result of the inability of the PUV fetus to release urine, given the incomplete or intermittent obstruction of the urethra. Notably, oligohydramnios results in poor development of the lung tissue and, when particularly severe, can lead to fetal death. Antenatal ultrasonography detects about 50% of PUV cases and has opened the way for fetal treatment [2]. With recent technological developments in the field of fetal-maternal medicine, vesicoamniotic shunting and fetal cystoscopy are now well-standardized in-utero procedures. However, the real benefit of vesicoamniotic shunting for PUV treatment has not yet been conclusively proven [3,4].

Therefore, postnatal primary valve ablation remains the gold standard treatment of PUV, and should be preferred to urinary diversion procedures, such as vesicostomy and ureterostomy, even in premature babies [5,6,7]. Depending on the degree of oligohydramnios, prematurity and respiratory distress secondary to congenital pulmonary hypoplasia may afflict PUV neonates, especially if oligohydramnios occurs during the first and second trimester of pregnancy. As such, there may be medical issues requiring immediate attention at birth that may cause delay in definitive surgical care of PUV. Herein, we describe three premature PUV babies presenting with complicated medical conditions, in whom prolonged indwelling urethral catheterization, initially adopted as temporizing treatment for bladder decompression, allowed definitive PUV treatment. Box 1 provides a summary box offering a synopsis of what is already known on the debated topic and what this article adds to the literature.

Box 1Thumbnail sketch of what is already known on the debated topic and what present article adds to the literature.
**What is already known on this topic** 
Posterior urethral valves (PUV) are the commonest cause of bladder out-let obstruction in male neonates, and a common cause of chronic kidney disease;in an ill, premature neonate with PUV, critical medical issues, urethral size, and anesthetic risks may complicate primary surgical treatment;urinary catheter represents the least invasive device to overcome initial PUV obstruction and allow adequate bladder decompression.
**What this study adds to the literature** 
Prolonged indwelling urethral catheterization may lead to definitive treat-ment in PUV patients facing long neonatal intensive care unit stays;PUV resolution may result from a combination of direct mechanical trauma caused by placement or inadvertent removal of the urinary catheter, and erosion caused by lateral pressure exerted by prolonged indwelling urethral catheter drainage;non-operative treatment may be considered as a viable alternative thera-peutic option for PUV in unstable premature babies.



## 2. Case Presentation

### 2.1. Case 1

A 26-year-old primigravida was transferred from an outside facility at 28 weeks’ gestation for further evaluation of severe oligohydramnios, associated with ultrasonographic findings of fetal lower urinary tract obstruction and ascites. Repeat ultrasonography and a complimentary fetal magnetic resonance imaging (MRI) performed at our institution were highly suggestive of PUV associated with urinary ascites and a slightly hydronephrotic right kidney. At 29 weeks’ gestation, she required an emergency C-section for placental abruption. At birth, the male neonate, weighing 1650 g, developed respiratory distress requiring endotracheal intubation and mechanical ventilation. He also underwent ascitic tapping. Initial serum creatinine was 1.4 mg/dl. Urinary decompression was accomplished by a 6 Fr Foley catheter passed into the bladder with the aid of its stiffening stylet. During advancement of the catheter some resistance was encountered when the membranous urethra was reached. Such resistance was stronger than what usually is offered as the catheter traverses the external sphincter. By applying gentle pressure, the catheter was successfully passed into the bladder, its balloon was then inflated with sterile water and correct placement was confirmed by ultrasonography. Following extubation on day-of-life 4, voiding cysto-urethrography (VCUG) showed a trabeculated bladder with a paraureteral diverticulum, absence of vesicoureteral reflux, and dilated posterior urethra. Because of some spillage of contrast medium into the peritoneal cavity, the catheter was kept in situ awaiting clinical improvement prior to proceeding with valve ablation. VCUG repeated on day-of-life 23, documented complete resolution of urinary extravasation and a less dilated posterior urethra. After catheter withdrawal, good urinary flow and negligible residual urine volume were documented during micturition, both findings highly indicative of valve disruption. Clinical course was complicated by Candidemia associated septic shock requiring aggressive therapy. He was eventually discharged home at 3 months of age. At 6 months of age, check cystoscopy performed using a 4.5/6.5 Fr compact cysto-urethroscope confirmed complete relief of bladder outlet obstruction. Static and dynamic renal scintigraphy showed normal function and excretion pattern of both kidneys, respectively. Repeat VCUG at 18 months disclosed further decrease of the posterior urethral dilatation, a less irregular bladder wall contour, and a small paraureteral diverticulum. He is now 19 years old, fully continent, with two normal kidneys, unremarkable renal function (serum creatinine 1.02 mg/dl, eGFR 106 mL/min/1.73m^2^), and without dysfunctional voiding symptoms (Figure 1). 

### 2.2. Case 2

A 26-year-old primigravida underwent fetal MRI at 29 weeks’ gestation for further characterization of urinary ascites associated with a keyhole sign and dilated, thick-walled bladder. The exam also documented bilateral hydronephrosis and mild lung hypoplasia. Subsequently, she developed progressive oligohydramnios and severe pre-eclampsia, requiring emergency C-section at 31 weeks’ gestation. At birth, the baby boy weighed 1945 g and required expeditious endotracheal intubation due to severe respiratory distress. Physical examination revealed a hugely distended bladder, palpable up to the umbilicus. Effective bladder decompression was achieved by an indwelling 6 Fr Foley catheter, which was inserted with some resistance felt during the passage into the posterior urethra. He received surfactant and was placed on nasal continuous positive airway pressure (CPAP) until day-of-life 4. Postnatal ultrasonography confirmed a thick-walled bladder, normal kidneys with a 10-mm bilateral hydronephrosis, and near-complete regression of ascites. VCUG was postponed, awaiting clinical improvement. On day-of-life 10, inadvertent urinary catheter removal was followed by a self-limiting urethral bleeding and easy bladder recatheterization. VCUG performed on day-of-life 18, showed a trabeculated bladder with posterior urethral dilatation. There was no reflux nor spillage of contrast medium into the peritoneal cavity. A good urethral stream was noticed, consistent with valves disruption. The baby continued to micturate smoothly and ultimately was discharge home on day-of-life 40. Check cystoscopy performed at 4 months of age documented valve remnants insignificant to bladder outflow. Follow-up ultrasonography showed progressive thinning of the bladder wall and normal kidneys. Renal scintigraphy performed at 3 years of age showed normal function and excretion pattern of both kidneys. He is now 9 years old, fully continent, with a 10-mm right hydronephrosis, negligible contour irregularity of the bladder wall, and minimal postvoid residual urine volume (Figure 2). Renal function is unremarkable (serum creatinine 0.67 mg/dl, eGFR 84 mL/min/1.73 m^2^), and there are no dysfunctional voiding symptoms. 

### 2.3. Case 3

A 34-year-old secundigravida, underwent fetal MRI at 30 weeks’ gestation confirming bilateral hydronephrosis, right dysplastic kidney, megacystis, and oligohydramnios. A male neonate, weighing 2.650 g, was delivered by elective C-section at 36 weeks’ gestation. Immediately after birth, he developed a mild pneumothorax with respiratory distress requiring nasal CPAP. Initial serum creatinine was 1.6 mg/dl. An initial attempt at transurethral catheterization using a 6 Fr Foley catheter was unsuccessful due to excessive resistance felt halfway through the urethra, and was followed by a self-limiting urethral bleeding. A 5 Fr umbilical catheter was therefore used to drain the bladder. Postnatal ultrasonography confirmed an enlarged and thick-walled bladder, associated with mild bilateral hydronephrosis, a dysplastic right kidney, and multiple small cysts with cortical subcapsular distribution in the contralateral left kidney. The umbilical catheter slipped out of the bladder after 72 h and was eventually replaced with a 6 Fr Foley catheter. On day-of-life 14, he was deemed fit enough to undergo VCUG. The test revealed a trabeculated bladder associated with a moderate posterior urethral dilatation and left-sided grade III vesicoureteral reflux. After catheter withdrawal, a good urinary stream and complete bladder emptying were documented during micturition (Figure 3). The baby was ultimately discharge home on day-of-life 17, with a relatively stable renal function. Repeat VCUG at 9 months disclosed a less irregular bladder contour, gradual shrinking of the posterior urethral dilatation, and reflux disappearance. The bladder emptied rapidly, and postvoid residual urine volume was small. He is now 2 years old and, following a first episode of urinary tract infection, had undergone second look cystoscopy that ruled out obstructing residual valves. Follow-up ultrasonography showed progressive thinning of the bladder wall and minimal postvoid residual urine volume (Figure 4).

Renal function remains stable with a serum creatinine of 0.53 mg/dl. However, eGFR (59 mL/min/1.73m^2^) falls within the range of stage 3 chronic kidney disease. 

## 3. Discussion

The ultimate goals of PUV treatment should include maximization of renal function, maintaining normal bladder function, minimizing morbidity, and preventing iatrogenic harm [8]. As such, affected infants should be treated soon after birth, even though their renal function outcome may be predetermined by genetics [9]. Initial PUV management entails transurethral catheterization, usually accomplished with a 6-Fr Foley catheter equipped with an internal semi-rigid stylet to facilitate the insertion. However, in very premature babies, smaller polyurethane tubes, originally designed for other purposes, may be required for bladder catheterization. In some instances, the urinary catheter dwell time may be prolonged because definitive PUV treatment needs to be delayed for other clinical issues demanding prompt medical attention, especially in vulnerable premature babies. This was the case of our patients, who, while awaiting clinical improvement experienced PUV resolution, likely resulting from a combination of direct mechanical trauma caused by placement or inadvertent removal of the urinary catheter, and erosion caused by lateral pressure exerted by prolonged indwelling urethral catheter drainage. Notably, the concept that pressure effects may be responsible for PUV disappearance is not new. Brandesky [10] first described such phenomenon in a PUV neonate in whom indwelling urethral catheter remained in place for 4 weeks, while awaiting to get rid of infection. VCUG showed a normal bladder without residual urine and the distal urethra being only slightly narrowed. The infant was discharged without any micturition difficulties. Unfortunately, the infant died 6 months later from aspiration pneumonia. Autopsy revealed a completely normal bladder and upper urinary tract. The bladder neck and posterior urethra showed a somewhat hypertrophic verumontanum and two mucosal folds representing residues of the original valves, causing no obstruction whatsoever. 

Interestingly, PUV resolution has also been described as result of spontaneous valves rupture occurring during fetal life. Such an exceptional event has been reported by Matsui et al. [11] in a 32-year-old woman with megacystis and posterior urethral dilatation detected at 15 weeks of gestational age. The megacystis, measuring 25 mm in longitudinal diameter, spontaneously disappeared a week later and did not reoccur throughout the rest of pregnancy. Assessment of amniotic fluid volume across gestation was unremarkable, and there were no indirect signs of bladder rupture, such as fetal urinary ascites or urinoma formation. Following a spontaneous vaginal birth at 37 weeks, the male newborn did not have difficulty in passing urine. However, ultrasonography revealed a thickened bladder wall with normal kidneys and upper urinary tracts. VCUG showed dilatation of the posterior urethra but confirmed normal bladder capacity with smooth voiding and no vesicoureteral reflux. Cystoscopy revealed remnant valves that were incised using a cold knife. At 7 months’ follow-up, the infant remained asymptomatic, and ultrasonography showed some improvement in bladder wall thickness.

In present cases, the antenatal suspicion of PUV was postnatally confirmed by VCUM in all. The exam documented the pathognomonic dilatation of the posterior urethra, and the relief of obstruction was ascertained during the voiding phase after removing the catheter [12]. Repeat VCUM during follow-up denoted further improvement in obstruction relief. Indeed, repeat VCUM allows for the comparative detection of any change in the degree of vesicoureteral reflux and posterior urethral dilatation [13]. However, the assessment of residual obstruction using VCUM may be sometimes not so clear cut [12]. As such, check cystoscopy is considered pivotal for directly establishing the persistence of residual bladder outlet obstruction, although a true diagnostic reference standard for urethral obstruction in boys does not seem to exist [14,15]. Additionally, present cases show that the endoscopic finding of residual valves does not necessarily imply inadequate relief of obstruction. Therefore, the routine use of check cystoscopy in the postoperative evaluation for residual valves may not be needed in all cases [13]. Diagnostic criteria of prenatal PUV include enlarged bladder, thickened bladder wall, posterior urethral dilatation, and oligohydramnios. Of these, increased bladder wall thickness and dilatation of the bladder have shown to be the most reliable diagnostic indicators [16]. Furthermore, these two parameters are also accurate, indirect indicators of obstruction relief. Nevertheless, VCUM and cystoscopy remain the key standards for both PUV diagnosis and documentation of relief of bladder outlet obstruction. Serial follow-up ultrasonography may allow an easy and reproducible assessment of progressive decrease in bladder wall thickness and postvoid residual volume [17,18,19,20]. Therefore, routine ultrasonography seems a safe first-line tool for postoperative assessment of PUV patients, and the use of more invasive investigations, such as repeat VCUM or second look cystoscopy, should be promptly considered when ultrasonographic measurements do not reach their normal values or if clinical symptoms occur.

Because of the retrospective nature of the present investigation, the possibility of selection bias cannot be discounted. However, we believe that spontaneous resolution of PUV as a result of valve disruption induced by insertion and prolonged duration of indwelling urethral catheter drainage is likely underreported. Nevertheless, we urge caution in generalizing present results and in the widespread adoption of this technique as a new standard of care.

## 4. Conclusions

Prolonged indwelling urethral catheterization, initially adopted as temporizing treatment for bladder decompression in PUV patients, may lead to definitive treatment and, therefore, may represent a viable alternative therapeutic option especially in unstable premature babies. However, check cystoscopy remains an essential and yet irreplaceable diagnostic tool to rule out persisting bladder outlet obstruction in PUV patients.

## Figures and Tables

**Figure 1 children-08-00408-f001:**
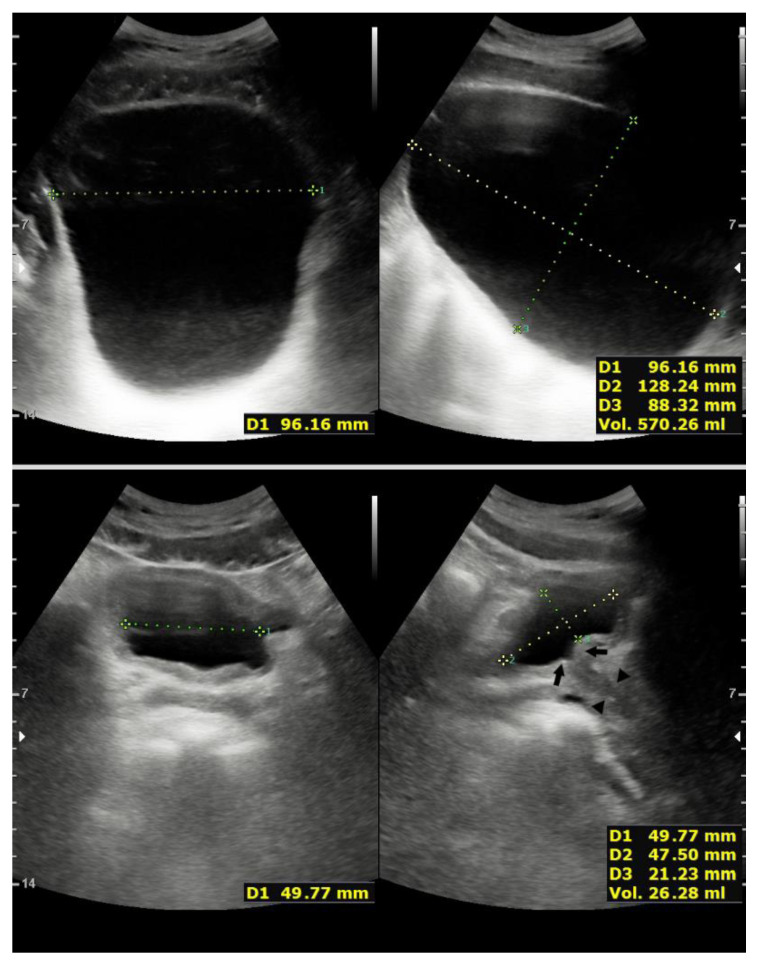
Top panels: Maximum bladder filling volume evaluated using direct ultrasonographic transabdominal bladder visualization. The greatest transverse (width), and anteroposterior (depth) and superior-inferior (height) distances are recorded, and volume measurement is automatically calculated and displayed on the machine’s screen. The bladder is smooth walled and with normal capacity. Bottom panels: Normal postvoid residual urine volume (<50 mL) evaluated shortly after a voluntary void. Note closure of the bladder neck (arrows) and proximal urethra (arrowheads) (lower right).

**Figure 2 children-08-00408-f002:**
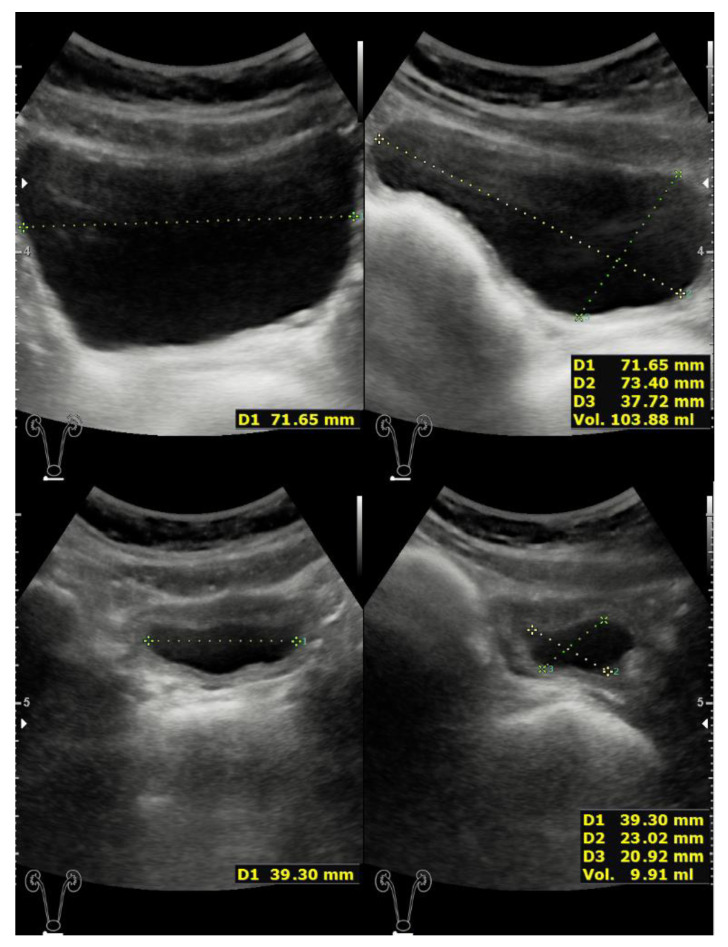
Top panels: Partially filled bladder with negligible bladder contour irregularity. Bottom panels: Insignificant postvoid residual urine volume.

**Figure 3 children-08-00408-f003:**
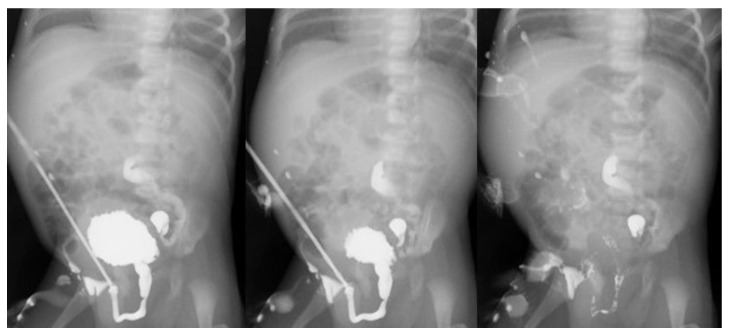
Serial radiograms of VCUM performed at 14 days of life, showing the voiding phase after catheter removal. Note a trabeculated bladder and a typically elongated and dilated prostatic urethra, both findings consistent with a diagnosis of PUV (**left**). A free voiding stream coming in a straight line (**left**,**middle**) and complete bladder emptying (**right**) were highly suggestive of urethral obstruction relief. Subsequent check cystoscopy documented valve disruption, likely induced by insertion and prolonged duration of the indwelling urethral catheter drainage.

**Figure 4 children-08-00408-f004:**
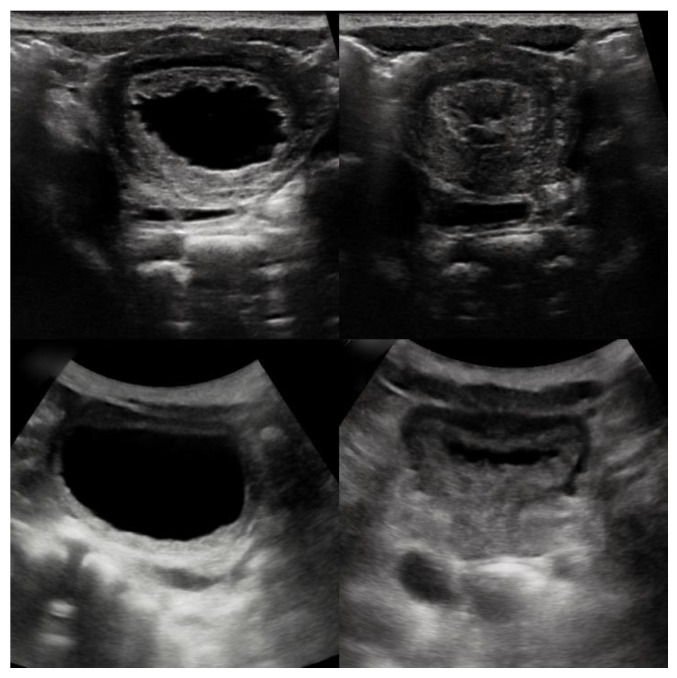
Top panels: Ultrasonographic transabdominal bladder visualization at 14 days of life. Partially filled bladder with a markedly thickened bladder wall and contour irregularity (**top left**). Absent postvoid residual urine volume evaluated shortly after a spontaneous micturition (**top right**). Note the walnut-sized bladder appearance secondary to the hypertrophic detrusor muscle. Bottom panels: Ultrasonographic transabdominal bladder visualization at 2 years of life. Note progressive thinning of the bladder wall and minimal postvoid residual urine volume (**lower right**).

## Data Availability

The data that support the findings of this study can be accessed by contacting the corresponding author upon reasonable request.

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
