# Peer review of "Prolonged Indwelling Urethral Catheterization as Minimally Invasive Approach for Definitive Treatment of Posterior Urethral Valves in Unstable Premature Babies"

_children, 2021, doi:10.3390/children8050408_

Round 1
Reviewer 1 Report
The authors presented three cases of premature children who underwent prolonged indwelling urethral catheterization as a definitive treatment of posterior urethral valves. After long follow up an unexpected good urinary flow and negligible residual urine volume were documented during micturition, suggestive of valve disruption induced by insertion and prolonged duration of indwelling urethral catheter drainage.
I am happy to saw this report because I noticed the same in several children we have treated on the same way. The report is well written, supported with adequate Figures. Several minor corrections should be performed as well as reorganization of discussion section prior to any favorable decision should be made:
- The authors used several abbreviations through the text without explanation (e.g. NICU, MRI…). For each abbreviation it is necessary to state the full title, at the place where it is first mentioned.
- Figures 1-4 should be divided in four subfigures (A, B, C and D) and each subfigure should be explained in the Legend of the Figure.
- Please state what cystoscope and what diameter of cystoscope was used in these children during cystoscopy.
- Discussion section needs to be re-written/re-arranged. Do not present a review of literature in this section. Do not discuss your findings piecemeal. Focus on results from main objectives of the study. Write in four sequential paragraphs (without headings); (i) summary (not data) of findings from present study; (ii) limitations of the study; (iii) logical and coherent comparison with existing literature with focus of comparison on main objective(s); and (iv) Implications for practice/policy/research with a concluding statement.
- Institutional Review Board Statement is missing. Even this was retrospective report according to new GDPR regulations IRB statement should be obtained.
- To emphasize the minimally invasive approach in unstable premature children I would suggest to the authors to change the title of the manuscript as follow: Prolonged Indwelling Urethral Catheterization as Minimally Invasive Approach for Definitive Treatment of Posterior Urethral Valves in Unstable Premature Babies.
Reviewer 2 Report
Thank you for considering me as a reviewer for this publication. I have provided my comments as follows.
General Comments:
In my opinion, this article is very well written and contributes to the existing knowledge about posterior urethral valves.
It can be accepted for publication but some spelling mistakes need to be corrected. I could not find any logical errors in the presentation and the approaches used.
Posterior urethral valves are not very often, but they are the most common congenital obstructive lesion of the urethra and a common cause of obstructive uropathy in infancy with serious consequences. The diagnosis can be established antenatally or postnatally. Patients who present after birth usually have less severe obstruction. Postnatally, treatment involves transurethral ablation of the offending valve. This method described by the authors is a good option in selected cases, and especially in children with a less complicated form of malformation in which definitive treatment can be delayed. In the discussion section, all components are critically reviewed.
This article will be useful for pediatricians, urologists, nephrologists, and other physicians that treat voiding disorders in children.
Specific Comments
- Introduction
The verb is does not agree with the subject. This is my suggestion:
Posterior urethral valve (PUV) is a rare congenital malformation...
or
Posterior urethral valves (PUV) are a rare congenital malformation...
- Discussion
It is necessary to move the second part of the sentence to the previous line
Such an exceptional event has been reported by Matsui et al. [11] in a 32-year-old woman with megacystis and posterior urethral dilatation detected at 15 weeks gestational age.
Round 2
Reviewer 1 Report
The authors have satisfactorily responded to all my questions and made the necessary changes to the manuscript. The manuscript has been improved and should be considered for publication in present form.